# Optimising the Spray Drying of Avocado Wastewater and Use of the Powder as a Food Preservative for Preventing Lipid Peroxidation

**DOI:** 10.3390/foods9091187

**Published:** 2020-08-27

**Authors:** Rahul Permal, Wee Leong Chang, Tony Chen, Brent Seale, Nazimah Hamid, Rothman Kam

**Affiliations:** 1Department of Food Science and Microbiology, Auckland University of Technology, Auckland 1010, New Zealand; rahul.permal@aut.ac.nz (R.P.); brent.seale@aut.ac.nz (B.S.); nazimah.hamid@aut.ac.nz (N.H.); 2Department of Biomedical Science, Auckland University of Technology, Auckland 1010, New Zealand; wee.leon.chang@aut.ac.nz; 3Department of Applied Science, Auckland University of Technology, Auckland 1010, New Zealand; tony.chen@aut.ac.nz

**Keywords:** avocado wastewater, spray drying, thiobarbituric acid reactive substance (TBARS), waste conversion, liquid chromatography–mass spectrometry (LC-MS), food preservation

## Abstract

Avocado wastewater (AWW) is the largest by-product of cold pressed avocado oil. The aim of this study was to valorise AWW by converting it into spray dried powder for use as a lipid peroxidation inhibiting food preservative. To increase the powder yield of AWW, addition of carriers and spray drying parameters (temperature and feed flow rate) were optimised. The highest AWW powder yield was 49%, and was obtained using 5% whey protein concentrate (WPC), with a feed flow rate of 5.8 g/min and an inlet drying temperature of 160 °C. The liquid chromatography mass spectrophotometry (LC-MS) analysis showed that AWW encapsulated with WPC had the highest retention of α-tocopherol (181.6 mg/kg powder). AWW with 5% WPC was tested as a preservative in pork fat cooked at 180 °C for 15 min. Thiobarbaturic acid reactive substances (TBARS) assay showed that the effectiveness of AWW powder was comparable to commercial additives such as butylated hydroxytoluene (BHT), butylated hydroxyanisole (BHA), and sodium erythorbate (E316).

## 1. Introduction

Demand for avocado (*Persea americana Mill.*) has significantly increased over recent years, due to its health promoting nutrients in the mesocarp. These include saturated, polyunsaturated, and monounsaturated fats; β-carotene; α-tocopherol; and essential minerals such as magnesium and potassium [1]. The five largest avocado producing countries, i.e., Mexico, Dominican Republic, Peru, Colombia and Indonesia, have collectively increased their avocado production from 2.4 million to 4.0 million tonnes from 2012 to 2018 [2].

Avocado fruit has applications in the cosmetics and food industries. The use of avocado oil for skin moisturisers and cosmetics have been reported since the 16th century. However, commercialisation of avocado oil for culinary purposes has only been popularised in the past 20 years [3]. Previous extraction methods of avocado oil for cosmetics and skin products have utilised harsh solvents technology such as hexane at high temperatures (>60 °C). The extracted oils would then be refined, bleached, and deodorised to remove any undesirable organoleptic properties [4,5]. Recent advancements in avocado oil extraction have included ultrasound treatment and supercritical CO_2_ methods [6,7]. Although successful and feasible, these technologies have yet to be commercialised.

The first successful development of cold pressed avocado oil (CPAO) was initiated in New Zealand and commercialised in 2000. CPAO appeals to consumers because of its attractive emerald colour, buttery flavour, and high smoking point of over 250 °C [4]. Consumers also benefit from its oleic acid content and presence of phytochemicals such as carotenoids, chlorophylls, and α-tocopherol. These phytochemicals act as antioxidants, which promote anti-inflammatory actions, regulate healthy blood lipid profiles, and increase bioavailability of fat soluble vitamins [8,9].

The CPAO process generates large masses of by-products in the form of seed, skin, pomace, and wastewater. Figure 1 depicts the major by-products generated from a typical CPAO production line. Using 1000 kg of avocado fruit during the early harvest season as a basis, 12.1% (*w*/*w*) and 15.3% of the input are removed as avocado seed and skin, respectively, during the de-stoning stage. Next, 15% pomace and 44.8% wastewater are removed from the malaxed avocado pulp emulsion at the three-phased decanting stage. The final by-product is removed through centrifugation as 5% residual water, producing a final 7.8% of pure CPAO [10].

Avocado by-products such as skin and seed can be turned into powders as nutrient-rich storable commodities, using convective drying procedures [11]. Furthermore, avocado seed has shown potential as a biofuel, an alternative source of starch, and a natural colour pigment [12,13,14]. However, valorisation of the major CPAO by-product that accounts for almost half the mass of avocado fruit input, avocado wastewater (AWW), is somewhat limited. AWW is a major concern in the avocado oil industry as it incurs high disposal costs and cannot be discarded into local drains due to its high organic matter. One way to circumvent this problem is to convert it into a higher value product.

Proximate analysis by Permal et al. (2019) [10] showed that AWW (% dry basis *w*/*w*) was primarily composed of 53.8 ± 9.4% lipids, followed by 22.2 ± 3.4% dietary fibres, 17.9 ± 0.6% ash, 10.3 ± 7.7% protein, and 0.9 ± 3.4% available carbohydrates. The authors successfully converted AWW from a commercial CPAO production line into dried powder with high antioxidants and total phenolic content. Furthermore, they successfully incorporated the powder into pork sausages as a preservative to prevent lipid peroxidation. However, the powder yields from this research were low, ranging from 18.6% to 32%, because the process was not optimised and no carriers were used. Researchers have found that products with certain sugars or high fat content could result in low powder yield during spray drying due to stickiness from sugar with low T_g_ (glass transition temperatures) or low melting point triglycerides [15,16,17]. To overcome this, gum, protein, or carbohydrate-based carriers can be added to encapsulate and form a barrier around freely dispersed active material. Microencapsulation through spray drying has proven to be efficient and practical, favouring product quality, increasing shelf life of fruit powders, maintaining stability of bioactive compounds, and increasing powder yield [16,18].

Previous research by Permal, Leong Chang, Seale, Hamid, and Kam [10] added spray dried AWW powder in sausages for inhibiting lipid peroxidation and found it to be as good as sodium erythorbate (E316). However, the bioactive component of AWW powder responsible for effective inhibition of lipid peroxidation was not determined in the study. Therefore, to add to the current body of literature, the aims of this study were to increase and optimise AWW powder yield through spray drying and to quantify the fat-soluble antioxidants responsible for preventing lipid peroxidation. Finally, the incorporation of this AWW powder into pure pork fat as a natural preservative was tested against commercial preservatives commonly used in the food industry.

## 2. Materials and Methods

### 2.1. Collection of Avocado Wastewater Samples and Fresh Avocado Fruits

Orangewood orchard in Northland, New Zealand supplied the Hass avocado fruits to Olivado Ltd. for commercial extraction of CPAO in late October 2019. The percentage of dry matter of these early season avocados was found to be 24%. The fruits were held at 20 °C inside wooden crates for ripening before processing. AWW was collected from the output of a three-phase decanter (Figure 1), on Olivado Ltd.’s CPAO processing line. Three batches of AWW was collected in 5 L PET bottles on three separate production days and immediately stored at 4 °C before spray drying. The fresh samples were freeze dried using the Alpha 1–2 LDplus Laboratory Freeze Dryer for 48 h, at −75 °C, and 1 × 10^−3^ mbar, and stored at −18 °C until further analysis.

### 2.2. Chemicals

Neocuproine, ammonium acetate, trolox (6-hydroxy-2,5,7,8-tetramethylchroman-2-carboxylic acid), propyl gallate, TBA (2-thiobarbituric acid), TEP (1,1,3,3-tetraethoxypropane), gallic acid, β-carotene, α-tocopherol, and butylated hydroxytoluene (BHT) were purchased from Sigma-Aldrich, Auckland, New Zealand. Copper (II) chloride dihydrate was purchased from VWR International, Aurora, CO, USA. Chloroform, methanol, hexane, and ethanol were purchased from Thermo Fisher Scientific, Auckland, New Zealand. Butylated hydroxyanisole (BHA) was purchased from BDH chemicals Ltd., Poole, England. Sodium erythorbate (E316) was obtained from D.M Dunningham Ltd., Auckland,New Zealand. Maltodextrin with a 10–12 dextrose equivalence (MD 10–12 DE) and MD 17–19 DE were both purchased from Davis Food Ingredients, New Zealand. Acacia gum was obtained from Hawkin Watts Ltd., Auckland, New Zealand. Lactose was purchased from BDH Chemicals and whey protein concentrate (WPC) sourced from Thompson’s, Auckland, New Zealand. Tri-chloroacetic acid (TCA) and Na_2_-EDTA (ethylenediaminetetraacetic acid disodium salt dehydrate) were purchased from Scharlau, Barcelona, Spain.

### 2.3. Spray Drying

AWW was spray dried using a laboratory scale Buchi mini spray dryer B-290 equipped with a Buchi B-296 dehumidifier (Switzerland) to remove all moisture in the spray drying air. The spray dryer unit was coupled with a 0.70 mm spraying nozzle. Preliminary tests found that atomising air at 49 m^3^/h, and an aspiration rate at 37 m^3^/h, were the most suitable parameters for optimal yield of AWW powder without carriers. To prepare each emulsion for drying, carriers were homogenised with AWW using the Silverson L4RT, at 5000 rpm, for 5 min. Four parameters were varied to find the ideal spray drying conditions for the highest yield of AWW powder. Firstly, the addition of different carriers including MD 10–12 DE, MD 17–19 DE, acacia gum (AG), lactose, and whey protein concentrate (WPC) was explored. The AWW feed rate was adjusted, varying between 3 to 11 g/min. The inlet temperatures were varied from 120 °C to 180 °C, at 20 °C increments, and carrier concentration of 1%, 5%, and 10% were used. All the parameters were collectively investigated for screening purposes to determine their effects on AWW powder yield using Equation (1) as described by Permal, Leong Chang, Seale, Hamid, and Kam [10]:(1)$$\mathrm{Yield}\;\%\;=\;\frac{Powder\;collected\;in\;cyclone\;\left(g\right)}{Mass\;of\;sample\;before\;spray\;drying\;\left(g\right)}\times \;100$$

### 2.4. Scanning Electron Microscopy (SEM) Imaging of Spray Dried Powders

Particle morphology for AWW powders using the five different carriers was evaluated as described by Permal, Leong Chang, Seale, Hamid, and Kam [10]. The particle size (μm) of SEM images was measured using ImageJ, an open source software (version 1.52u) developed at the National Institutes of Health (NIH), Bethesda, MD, USA.

### 2.5. Extraction and Liquid Chromatography Mass Spectrophotometry (LC-MS) Analysis of Acid β-Carotene and α-Tocopherol

The method to quantify β-carotene and α-tocopherol from AWW using liquid chromatography mass spectrophotometry (LC-MS) has not previously been studied, Therefore, the extraction method used in this research was specifically designed for this purpose. Briefly, 1 g of AWW powder was measured into 15 mL falcon tubes. Then, 0.5 mL of methanol followed by 1 mL hexane were added to the powder. Each solvent addition was vortexed for 30 s. The falcon tubes were, then, centrifuged for 5 min, at 2700 rpm, using the Vortex-Genie II. The resulting top layer of hexane was removed using a glass pipette and dispensed into a 5 mL glass test tube. The extraction using 1 mL hexane was repeated two more times. All the hexane was evaporated using nitrogen. Then, the reduced viscous solution was re-dissolved in 0.2 mL of ethanol, vortexed for 30 s, centrifuged for 5 min at 2700 RPM, and transferred into silanised inserts. The inserts were capped inside 1.5 mL amber vials and immediately analysed in the LC-MS.

Quantification was performed by using commercial β-carotene and α-tocopherol standards to generate a calibration curve in the range of 0.16–20 mg/L (R^2^ = 0.999). Chromatographic repeatability (*n* = 10) was estimated and the residual standard deviation (RSD) was calculated at 6% for α-tocopherol and 11% for β-carotene. Recovery for α-tocopherol and β-carotene was calculated using Equation (2). The recovery experiment was carried out by spiking 0.1 g of powder with 50 uL of 10 mg/L standards and underwent the same extraction method for samples as described above.
(2)$$\mathrm{Recovery}\;\%\;=\;\frac{LC\;MS\;concentration\;\left(\frac{mg}{L}\right)\;reading}{Theoretical\;vitamin\;concentration\;\left(\frac{mg}{L}\right)}\times \;100$$
(3)$$\mathrm{LoD}\;=\;3.3\;\times \;\frac{S_{\mathrm{residuals}}}{\mathrm{Slope}}$$
(4)$$\mathrm{LoQ}=10\;\times \;\frac{S_{residuals}}{Slope}$$

Limit of detection (LoD) and limit of quantification (LoQ) were calculated based on Equations (3) and (4) [19]. S_residuals_, is the residual standard deviation from the calibration curve of compounds in the LoD region, and slope is the slope from the calibration curve of each component. LoD and LoQ for α-tocopherol were 0.04 mg/L and 0.13 mg/L, respectively, and for β-carotene, LoD and LoQ were 0.39 mg/L and 1.30 mg/L, respectively.

The LC-MS analysis was conducted using an Agilent 1260 Infinity Quarternary LC System (Santa Clara, CA 95051, USA). The system consisted of the following components: 1260 infinity quaternary pump, 1260 infinity ALS sampler, 1260 infinity TCC column component and a 1260 infinity diode array detector (DAD) connected to a 6420 triple quadrupole LC/MS system with multimode ionisation source. A Waters XSelect CSH C18 (2.1 × 100 mm, 3 µm) column was used for the analysis. The mobile phases were composed of water containing 0.1% (*v*/*v*) formic acid (A) and methanol (B). The initial gradient condition was 5:95 (A:B). From 0 to 1 min, B was increased to 100% and held for 7.5 min, then, from 8.5 to 9.2 min, B was decreased to 95%. The injection volume was 3 µL and the total run time was 15 min for each sample.

The MS ionisation source conditions were set as outlined: Capillary voltage of 2 kV and corona current of 4 µA, drying gas (N_2_) temperature of 300 °C at a flow rate of 5 L/min, vaporiser temperature of 250 °C, and nebuliser pressure of 50 psi were used. The positive ion mode was performed with MRM for quantitative analysis. Precursor-to-product ion transition used for α-tocopherol was, [M+H]^+^
*m*/*z* 431 → 165,137 with a fragmentor voltage of 100 V and collision energy of 32 eV and 50 eV, respectively. The precursor-to-product ion transition used for β-carotene was [M+H]+ *m*/*z* 537 → 537, 277 with a fragmentor voltage of 160 V and collision energy of 1 eV and 16 eV, respectively.

### 2.6. Preparation of Pork Fat for Lipid Peroxidation Tests

Pork fat was purchased from a local butcher in Auckland, New Zealand and used to quantify the absolute degree of fat peroxidation using the TBARS (thiobarbaturic acid reactive substances) test. Pork fat was used instead of protein-rich meat, as research has shown that proteins can interfere with TBARS test by giving higher MDA (malondialdehyde) values [20,21,22]. To prepare the samples, pork fat was minced using a Kenwood Pro 1400 mincer, and then equally divided into seven treatments. Treatment 1 was used as a control with no additives and Treatment 2 contained 0.04% (*w*/*w*) sodium erythorbate (E316). FSANZ [23] states that 0.04% (*w*/*w*) of E316 is the maximum allowable limit to be added into meats. Therefore, Treatments 3, 4, 5, and 6 were based on the cupric ion reducing antioxidant capacity (CUPRAC) equivalence of each additive to E316. Treatments 3, 4, 5, and 6 contained 1.5% (*w*/*w*) AWW, 0.1% (*w*/*w*) BHT (butylated hydroxytoluene), 0.01% (*w*/*w*) BHA (butylated hydroxyanisole), and 1.86% (*w*/*w*) α-tocopherol respectively. Treatment 7 was used as a positive control and contained 0.1% (*w*/*w*) WPC. Once prepared, the samples were transferred into glass beakers in triplicate and baked at 180 °C for 15 min using a Piron PF4005D oven (Italy). Then, all samples were cooled to room temperature and immediately analysed for degree of fat oxidation using the TBARS protocol.

### 2.7. Cupric Ion Reducing Antioxidant Capacity (CUPRAC) Analysis for Equivalence of Antioxidant Activity amongst Selected Preservatives

The extraction of antioxidants from samples was carried out as reported by [10]. Cupric ion reducing antioxidant capacity (CUPRAC) assay was conducted as detailed by [24]. Firstly, 1 mL of appropriately diluted sample, was added into 1 mL of CuCl_2_·2H_2_O (0.01 M), NH_4_AC buffer (1 M, adjusted to pH 7), neocuproine (0.075 M), and then 0.1 mL of distilled H_2_O (total volume of 4.1 mL). The sample solutions were held for 5 m at room temperature and absorbance was measured against a reagent blank (1 mL, neocuproine, CuCl_2_·2H_2_O, NH_4_AC buffer solution, and 1.1 mL water) using a GE ultrospec 7000 spectrophotometer at 450 nm. Concentration was calculated from a Trolox standard curve (5 to 170 mg/L, R^2^ = 0.9965).

The CUPRAC analysis (Table 4) indicated that the antioxidant capacity of β-carotene was not significantly different from the AWW powder (*p* > 0.05). However, preliminary TBARS analysis of β-carotene produced MDA values far higher (~6 mg MDA/kg of pork fat) than the control. This could have been attributed to its intense orange pigment interfering with spectrophotometry results. Therefore, β-carotene was not presented in Figure 4.

### 2.8. Lipid Oxidation of Pork Fat Using TBARS

The Thiobtrbaturic acid reactive substances (TBARS) value was determined colorimetrically [10]. Therefore, 1 g of pork fat sample was measured out in triplicate, inside 15 mL falcon tube, and mixed with a 5 mL solution containing 7.5% trichloroacetic acid solution (TCA), 0.1% propyl gallate, and 0.1% EDTA-Na_2_, along with 5 mL of TBA reagent (0.02 M thiobarbituric acid in distilled water). The tubes were incubated at 100 °C for 40 min and cooled to ambient room temperature in an ice bath. Once cooled, the absorbance was measured at 532 and 600 nm. The extent of peroxidation in terms of malondialdehyde equivalents was determined based on a series of TEP (1,1,3,3-tetraethoxypropane) standards (R^2^ = 0.9996).

### 2.9. Statistical Analysis

Samples were analysed in triplicate and data were expressed as mean ± standard deviation. One-way analysis of variance (ANOVA) with Tukey pairwise comparison of means was performed using the XLSTAT software (version 2018.7). A difference of *p* ≤ 0.05 was considered to be significant.

## 3. Results and Discussion

### 3.1. Optimising Spray Drying Conditions

Previous research by Permal, Leong Chang, Seale, Hamid, and Kam [10] on spray dried AWW, exhibited low yields not exceeding 32%, because of powder sticking to the spray drier chamber wall. Proximate analysis from this study revealed that AWW was high in lipid content (53.8 ± 9.4% *w*/*w* dry basis), which could result in stickiness due to the presence of low melting point triglycerides [15]. One way to circumvent this issue was to spray dry AWW using dehumidified air as a drying medium and to encapsulate the avocado oil present in AWW using carriers such as MD 10–12 DE, MD 17–19 DE, AG, lactose, or WPC. Other spray drying parameters including, feed flow rate (g/min), temperature, and carrier concentration were also varied to optimise AWW powder yield.

The optimising process started with the selection of the best encapsulating carrier for AWW during spray drying. Figure 2A shows that the WPC carrier gave the highest average powder yield at 48% followed by lactose at 32%. There were no significant differences (*p* > 0.05) between MD 10–12 DE, MD 17–19 DE, and AG as compared with the control with yields ranging from 19 to 24%. Therefore, WPC was chosen as the carrier of choice for the consecutive spray drying experiments.

Altering feed flow rate (g/min), as shown in Figure 2B, had a significant impact on powder yield (*p* < 0.05) when 5% WPC loading and 140 °C drying temperature were kept constant. The highest average yield was observed at a feed flow rate of 5.8 g/min. Further increase in feed flow resulted in decreased yields. Spray drying of AWW with 5% WPC resulted in an increase in powder yield with an incremental increase of spray drying inlet temperatures (Figure 2C). The highest average yield of 49% AWW powder was obtained at inlet temperatures of 160 °C and 180 °C. As shown in Figure 2D, carrier concentration of 5% WPC resulted in significantly higher (*p* < 0.05) yield than 1% WPC (32%) and 10% WPC (43%).

In this study, WPC resulted in the highest powder yield as compared with other carriers. Proteins have many desirable functional properties as a wall material, including, solubility, ability to interact with water, film formation, emulsification, and stabilization of emulsion droplets. Proteins are effective at encapsulating oils, as they change their structure during emulsification through unfolding and adsorption at the oil water interface. These proteins form resistant multilayers around oil droplets and with the help of repulsive forces they make stable emulsions that are critical for encapsulation purposes [25]. Bae and Lee [17] reported that WPI (whey protein isolate) was more effective in encapsulating CPAO when used alone as compared with the incorporation with maltodextrin at a 1:1 ratio. Similarly, Jimenez et al. [26] reported WPC as a good encapsulating agent for CLA (conjugated linoleic acid). Their study indicated a lower degree of CLA oxidation using WPC as compared with WPC/MD (1:1 ratio of WPC and maltodextrin) or GA (gum Arabic). Du et al. [27] compared the efficiency of five different carriers to overcome stickiness tendencies of persimmon pulp powder. The study concluded that powder yield with addition of 25% WPC was equivalent to the yield of using 45% MD 14–16 DE, 30% GA, and 30% starch sodium octenyl succinate (SSOS). Comparable results were also reported by Fang and Bhandari [28] who used maltodextrin and whey protein isolate (WPI) as carriers for spray drying bayberry juice. This study found that 1% WPI was sufficient for powder recovery (>50%), whereas carrier loading of more than 30% of maltodextrin was required for the same purpose. Du, Ge, Xu, Zou, Zhang, and Li [27] attributed protein as an effective carrier agent due to the surface activity of protein. The surface activity of a protein allows it to transfer through the interface of a feed solution and rapidly form a film while drying. This film effectively overcomes bonding of consecutive droplets and the stickiness between droplets to the chamber wall.

Figure 2B shows a gradual decrease in AWW powder yield with an increasing feed flow rate. Khalilian, Movahhed, and Mohebbi [29] explained that increasing feed flow rate could produce dried samples with higher moisture content as larger feed volumes were being passed over at the same time period. Therefore, plasticisers, such as water (T_g_ = −135 °C), decrease the T_g_ of atomised feed material, leading to insufficient drying of the powder and lower powder yield [15].

Figure 2C shows that increasing inlet temperature consequently increases AWW powder yield. Similarly, Fazae et al. [30] reported the positive effects of increasing inlet temperatures (110 °C to 150 °C) on the output of mulberry powder. The authors attributed this higher yield to the greater efficiency for heat and mass transfer processes. Increasing inlet temperatures decreased the chance of inadequately dried powders attaching onto the spray dryer’s chamber wall. In contrast, Dantas, Pasquali, Cavalcanti-Mata, Duarte, and Lisboa [18] concluded that utilising a lower inlet temperature (80 °C) in conjunction with adjusted atomization flow rates, produced higher yields for avocado powder. Dantas, Pasquali, Cavalcanti-Mata, Duarte, and Lisboa [18] utilised inlet temperatures on the lower spectrum (80–120 °C) as their feed material (avocado flesh and milk) contained considerable amounts of low molecular weight sugars with low T_g_, as well as high amounts of fats. Dantas, Pasquali, Cavalcanti-Mata, Duarte, and Lisboa [18] found that application of inlet temperatures above 80 °C would have resulted in lower powder yield as a result of stickiness with drying temperatures coming closer to glass transition temperatures.

The improved yield increasing from 1 to 5% WPC (32% to 49%, respectively), as shown in Figure 2D, was consistent with studies by both Fazaeli, Emam-Djomeh, Kalbasi Ashtari, and Omid [30] and Dantas, Pasquali, Cavalcanti-Mata, Duarte, and Lisboa [18]. Both studies found that increasing carrier concentrations, significantly increased process yield of black mulberry juice and avocado powder, respectively. The authors reported that increasing carrier concentration increased T_g_ values of other amorphous fractions present in the mixture which were naturally low in T_g_ components. Tonon et al. [31] demonstrated that increasing carrier concentration decreased process yield, which could be due to an increase in mixture viscosity. This may explain the decreased yield in this study when using 10% WPC to encapsulate AWW as compared with 5% WPC.

### 3.2. Powder Morphology for Spray Dried AWW Powder

AWW powder samples in Figure 3B–F were produced with a feed flow rate of 5.8 g/min, 5% (w/w) carrier loading, and spray drying inlet temperature of 140 °C. The SEM analysis of spray dried AWW encapsulated with MD 10–12 DE (4.3 ± 2.36 μm), MD 17–19 DE (4.6 ± 1.75 μm), AG (4.6 ± 3.39 μm), lactose (4.14 ± 1.41 μm), and WPC (4.4 ± 2.23 μm) showed little difference in average particle size. Morphology of freeze-dried AWW (Figure 3A) was the most unique as compared with those spray dried with carriers. The powder appeared as, agglomerated, irregular flake particles. The highest degree of polymerisation of encapsulated powders was seen in lactose (Figure 3E) because of its low T_g_, 101 °C [15]. However, AG, MD 10–12 DE, MD 17–19 DE, and WPC were in a mass of spherical agglomerates.

Previous SEM micrographs of spray dried AWW powder without encapsulation appeared as agglomerates of smaller particle sizes (<4 μm), rather than separate individual components [10]. In the present study, increasing inlet temperature resulted in a lower degree of coalescence, thus, producing distinguishable spherical particles. With the addition of carriers, spray dried AWW particles still appeared to be agglomerated, but the degree of particle separation and spherical morphology were higher. There was low particle size variability between MD 10–12 DE and MD 17–19 DE (Figure 3B,C). However, Tonon et al. [32] found that açai powder particles produced with MD 10 DE exhibited a median diameter higher than MD 20 DE and AG. They reported that the increase in particle size was influenced by the molecular size of each carrier agent.

Spray dried AWW powder had a high degree of particle agglomeration, making it difficult to distinguish individual particle sizes (Figure 3B–F), possibly due to the presence of surface fat bridging between particles. Surface topology of powders, as shown in Figure 3B–F, indicated the presence of some pores, cracks, and surface depression. Kim et al. [33] explained that quick formation of powder crusts could cause surface fissures or breakages. Alternatively, if the microcapsule crust is moist and supple, the particle could deflate and shrivel upon cooling.

### 3.3. Quantifying Total β-Carotene, α-Tocopherol Content in AWW Powder Using the LC-MS

In a previous study, spray dried AWW powder was found to be rich in antioxidants, more so than freeze-dried avocado flesh [10]. The powder was successfully added into pork sausages as a natural preservative and showed no significant differences in terms of lipid peroxidation as compared with use of the synthetic E316 (sodium erythorbate) preservative. Analysis of antioxidants were conducted spectrophotometrically providing total antioxidant capacity and activity. However, research from Permal, Leong Chang, Seale, Hamid, and Kam [10] did not identify compounds contributing towards the inhibition of lipid peroxidation. Wong et al. [34] stated that CPAO contained high concentrations of naturally occurring antioxidants, specifically α-tocopherol (70–190 mg kg^−1^ oil) and carotenoids (1.0–3.5 mg kg^−1^ oil). Yang et al. [35] reported that not all avocado oil droplets were removed from AWW during CPAO production, which suggested that the preservative properties of AWW could be largely influenced by the presence of carotenoids and tocopherol.

The LC-MS analysis carried out in this study focused on total β-carotene and α-tocopherol content in AWW powder because of their solubility in lipids. This is the first time such an analysis has been conducted on spray dried AWW. The current extraction method was relatively efficient in recovering α-tocopherol (93.4%) (Table 1). The powder samples which were analysed included those spray dried with carriers, freeze-dried avocado flesh, and neat AWW. As shown in Table 2, avocado flesh had the highest concentration of α-tocopherol (278.7 ± 11.76 mg α-tocopherol/kg powder), nearly triple the amount of what was present in freeze-dried AWW. Interestingly, freeze-dried AWW showed no statistical difference to AWW powders encapsulated with lactose, acacia gum, MD 10–12 DE, and MD 17–19 DE (*p* > 0.05). AWW encapsulated with WPC was the only spray dried sample that had significantly higher α-tocopherol content as compared with freeze-dried AWW. This demonstrated the ability of WPC to maintain and protect α-tocopherol from degradation. Powders dried at 110–160 °C (Table 3) without the addition of carriers exhibited a higher concentration of α-tocopherol than encapsulated samples (Table 2). This is due to the concentration effect of carriers, as addition of carriers would have increased solid content of the sample without increasing phytochemical content.

Spray drying temperatures from 110 °C to 150 °C showed no statistical differences in terms of α-tocopherol content (*p* > 0.05) of AWW powder. Furthermore, the highest inlet temperature of 160 °C produced AWW powder with a significantly higher (*p* < 0.05) α-tocopherol content (320.2 ± 51.09 mg α-tocopherol/kg powder) than AWW powders spray dried at 140 °C and 150 °C WPC (186.3 ± 44.77 and 196.9 ± 23.7 mg α-tocopherol/kg powder, respectively). However, due to the high variability in α-tocopherol content, results utilising an inlet temperature of 160 °C was not significantly different from 110 °C, 120 °C, and 130 °C treatments (187.9 ± 23.09, 226.8 ± 23.44, 201.0 ± 33.57 mg α-tocopherol/kg powder, respectively).

The LC-MS recovery protocol for β-carotene was relatively low (18%, Table 1) which consequently produced inconsistent data for total β-carotene in AWW powder, as shown in Table 2 and Table 3. Nonetheless, the LC-MS analysis in Table 3 shows no statistical differences for β-carotene content in powders dried between 110–160 °C. Table 2 shows that WPC retained a significantly higher (*p* < 0.05) concentration of β-carotene (15.1 ± 0.23 mg β-carotene/kg powder) as compared with AWW encapsulated with lactose, acacia gum, MD 10–12 DE, and MD 17–19 DE (0.0, 0.7 ± NA, 0.0, 1.5 ± NA mg β-carotene/kg powder). The results showed that WPC was the most efficient carrier for retaining both α-tocopherol and β-carotene content in AWW powder. Several reports have suggested that protein was an ideal carrier for preserving nutraceutical components of powders due to its excellent film forming ability. The high recovery of powder and α-tocopherol retention with only 5% of WPC could be due to surface active properties of WPC in solutions. Fang and Bhandari [28] found that proteins migrated towards the air and water interphase, reduced surface fat composition of droplet particles, and consequently decreased exposure of the oils to the temperature extremities in the drying chamber. Secondly, the migration of protein onto the surface of sprayed dried powder was capable of rapidly forming a very thin protein rich film. This protein film could have a higher T_g,_ allowing it to remain in the glassy state and protecting the oils from attaching onto the hotter surfaces of the spray dryer. Moreover, a study by Dian et al. [36] found that maltodextrin and sodium caseinate retained higher levels of carotene in encapsulated palm oil than a blend of maltodextrin and acacia gum.

Utilising the highest spray drying temperature of 160 °C did not degrade α-tocopherol concentration in AWW powders without carriers. Similarly, Permal, Leong Chang, Seale, Hamid, and Kam [10], reported that increasing the spray drying inlet temperatures (110–160 °C) for AWW without carriers either increased or maintained the level antioxidant activity for AWW. Sabliov et al. [37] reported that α-tocopherol was stable at high temperatures in the absence of oxygen, but under normal atmospheric conditions the rate of degradation for α-tocopherol increased with increasing temperatures. They showed a reduction in α-tocopherol under atmospheric pressure at temperatures ranging from 40 to 180 °C and concluded that increasing temperature not only decreased α-tocopherol content but also increased the rate of decrease, i.e., 55% of α-tocopherol had degraded after 2 h at 180 °C, and almost 80% of free α-tocopherol reduced after 5 h of exposure. There was no difference observed at temperatures below 120 °C, whereby only 30% of free α-tocopherol degraded after 5 h. Contrary to holding times reported for α-tocopherol degradation, spray drying is an efficient system for delivering powders straight into the collection vessel without being held at high inlet temperatures for a long time. Hence, the degradation of α-tocopherol should be minimal. Even so, this increase in α-tocopherol could be a result of rapid crust formation at the surface of the powder, forming a barrier to protect the α-tocopherol rich oils at the particle center [10,17].

Similar to the stability of α-tocopherol, altering the spray drying inlet temperature did not have a significant impact (*p >* 0.05) on the concentration of β-carotene in spray dried AWW. In contrast, Khalilian, Movahhed, and Mohebbi [29] reported a sharp decrease in β-carotene content for carrot celery juice when increasing spray drying inlet temperatures (120–170 °C). Interestingly, the authors reported that increasing carrier loading (10% to 30% maltodextrin) decreased β-carotene content. However, the addition of 5% WPC as a carrier in the current study significantly increased retention of β-carotene (15.1 ± 0.23 mg β-carotene/kg powder, *p* < 0.05) as compared wirh spray dried powder without carriers (0.4 mg β-carotene/kg powder at 150 °C). Compared to lactose, AG, MD 10–12 DE, MD 17–19 DE, and AWW powders without encapsulation, WPC’s rapid film forming capabilities provided superior protection for β-carotene.

### 3.4. Inhibition of Lipid Peroxidation Using AWW Powder

Lipid peroxidation is a major cause of deterioration in fat rich foods, especially those containing polyunsaturated fats (PUFAs). Sodium erythorbate, BHT, BHA, and α-tocopherol are common lipid soluble antioxidants used by food industries to prevent lipid oxidation in foods. The CUPRAC assay was carried out on four other synthetic additives to match their antioxidant effect to E316. These included β-carotene, BHT, BHA, and α-tocopherol. Table 4 shows the mg Trolox equivalent/100 g of powder for each additive, as well as their equivalent concentration needed to produce an antioxidant capacity, similar to 0.04% of E316. The antioxidant capacity of BHA was significantly higher (*p* < 0.05) as compared with the other six additives and approximately 150 times more potent than AWW powder. Figure 4 shows that adding 1.50% AWW was just as effective as 0.04% E316, 0.10% BHT, 0.01% BHA, and 0.20% α-tocopherol in inhibiting lipid peroxidation. There were no statistical differences in mg MDA/kg of pork fat after cooking the fat with different antioxidants at 180 °C for 15 min. Interestingly, α-tocopherol was not significantly different from the control (*p* > 0.05) with respect to preventing lipid oxidation. Permal, Leong Chang, Seale, Hamid, and Kam [10] reported that AWW powder was an alternate antioxidant additive, effective in preventing fat oxidation in pork sausages. The study also found that there were no significant differences in levels of MDA complexes formed using 0.20% *w*/*w* AWW without carriers and 0.04% E316 in the sausages. In contrast, Table 4 shows a higher equivalent of AWW (1.5% *w*/*w*) required to match E316 (0.04 *w*/*w*). The higher concentration of AWW is likely due to the concentration effect of WPC, increasing non-phytochemical concentration of AWW powder produced in the current study.

Of the four control antioxidants studied, pure α-tocopherol was not as effective in preventing lipid oxidation. Ottaway [38] explained that, α-tocopherol was readily oxidised by air. It is heat-stable in the absence of oxygen, but if heated in the presence of oxygen it will degrade faster. The results of this study showed that WPC was able to protect α-tocopherol from degradation during high temperature processing of AWW. The encapsulation of AWW using WPC could preserve the effectiveness of α-tocopherol as an antioxidant, as shown in Table 2. However, previous research has shown that proteins could interfere with the TBARS assay [20,21,22]. Therefore, the WPC powder which was used as a positive control for the TBARS test was shown to increase MDA content (Figure 4). However, when AWW was encapsulated with WPC, there was no interference with MDA formation due to the low concentration of WPC in AWW as compared with the positive control.

## 4. Conclusions

The design of this study was aimed at optimising existing spray drying conditions of AWW to increase AWW powder yield. The highest yield (49%) was obtained using 5% WPC as a carrier, feed flow rate of 5.8 g/min, and an inlet temperature of 160 °C. The SEM images revealed that the addition of carriers was beneficial in terms of powder morphology. There was a lower degree of particle agglomeration with encapsulation. Encapsulation using WPC was efficient in protecting α-tocopherol and β-carotene. The addition of spray dried AWW into cooked pork fat was as effective as synthetic additives such as BHT, BHA, E316, and α-tocopherol in preventing lipid oxidation.

## Figures and Tables

**Figure 1 foods-09-01187-f001:**
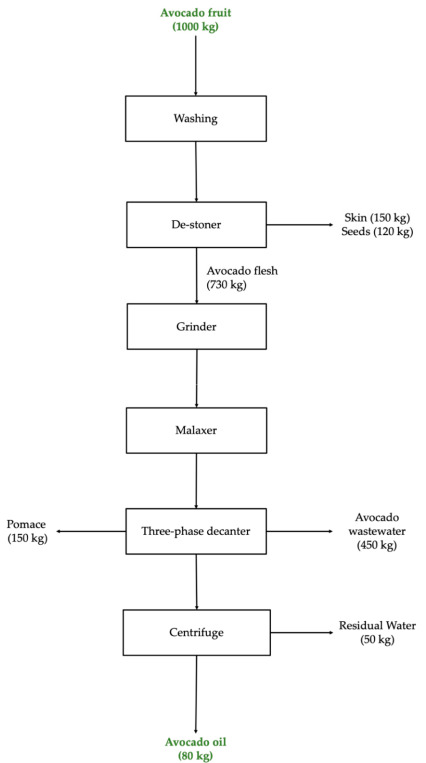
Process flow diagram of a typical commercial cold pressed avocado oil (CPAO) extraction process. All the avocado waste output values from the early harvest season were obtained using an input of 1000 kg avocado fruits as the basis.

**Figure 2 foods-09-01187-f002:**
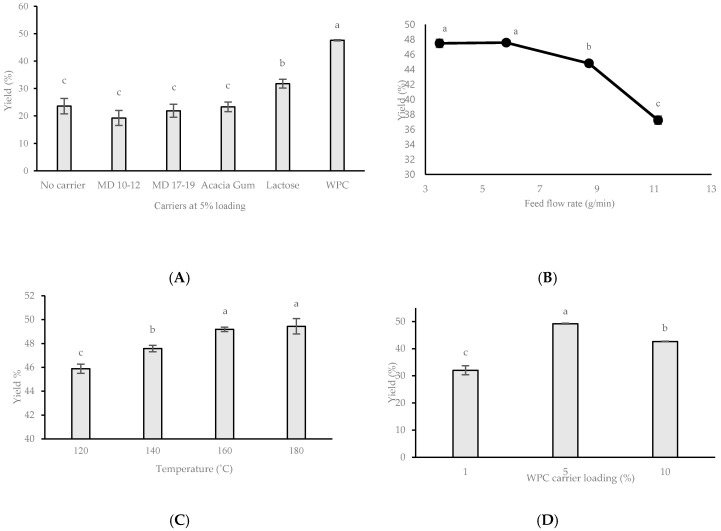
Yield of spray dried avocado wastewater (AWW) under different spray drying conditions. Each data point was measured in triplicate (*n* = 3) where error bars represent standard deviation of means. (**A**) Effect of carrier types on powder yield when drying temperature of 140 °C, 5.8 g/min feed flow rate, and carrier concentration of 5% w/w with respect to AWW were kept constant; (**B**) Effect of feed flow rate on powder yield when drying temperature of 140 °C and 5% whey protein concentrate (WPC) carrier loading were kept constant; (**C**) Effect of spray drying temperature on powder yield when flow rate of 5.8 g/min and 5% WPC carrier loading were kept constant; (**D**) Effect of WPC carrier loading at a constant feed flow rate of 5.8 g/min and drying temperature of 160 °C. Superscript letters that are the same do not differ statistically based on Tukey’s test (*p* < 0.05).

**Figure 3 foods-09-01187-f003:**
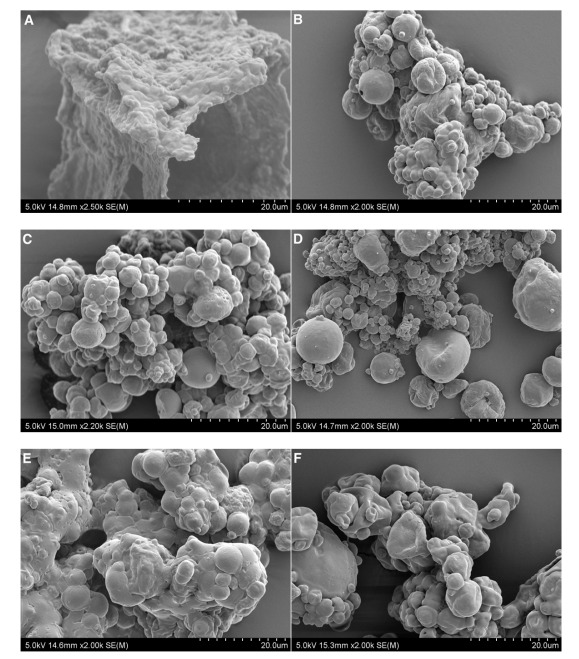
SEM images of AWW powders. The following data represents powder treatment as well as mean particle size ± standard deviation of powder. (**A**) Freeze-dried AWW, 1.57 ± 0.29 μm; (**B**) Maltodextrin with a 10–12 dextrose equivalence (MD 10–12 DE), 4.3 ± 2.36 μm; (**C**) MD 17–19 DE, 4.6 ± 1.75 μm; (**D**) Acacia gum, 4.6 ± 3.39 μm; (**E**) Lactose, 4.14 ± 1.41 μm; (**F**) WPC, 4.4 ± 2.23 μm. Samples were randomised images of what was seen under the SEM for the various parameters. Samples (**B**) to (**F**) were subjected to spray drying conditions of 140 °C, 5.8 g/min feed flow rate, and carrier concentration of 5% *w*/*w* with respect to AWW.

**Figure 4 foods-09-01187-f004:**
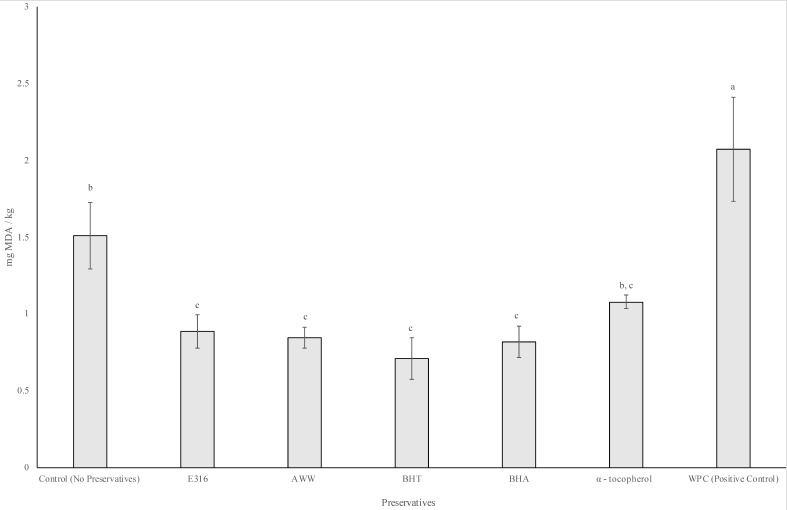
Thiobarbaturic acid reactive substances (TBARS) values (mg MDA/kg) in cooked pork fat containing various synthetic and commercially available additives and spray dried AWW powder. Superscript letters that are the same indicate that mean values do not differ statistically when using the Tukey’s test (*p* < 0.05). The analysis of each sample was conducted in triplicate. Error bars represent the standard deviation of means. AWW powder used was spray dried at 160 °C at a flow rate of 5.8 g/min and encapsulated with 5% WPC.

**Table 1 foods-09-01187-t001:** LC-MS recovery of α-tocopherol and β-carotene.

Active Component	Initial Concentration (µg/g)	^1^ Amount Spiked (µg)	Expected LC-MS Reading (µg/g)	Actual LC-MS Reading (µg/g)	^2^ Recovery (%)
α-Tocopherol	1.0	0.5	1.5	1.4	93.5 ± 3.1
β-Carotene	0.2	0.5	0.7	0.1	18.0 ± 8.5

^1^ Concentration of spiked α-tocopherol and β-carotene. ^2^ Data is mean ± standard deviation (*n* = 3).

**Table 2 foods-09-01187-t002:** Total α-tocopherol and β-carotene content of freeze-dried and spray dried AWW and avocado flesh.

Samples	mg α-Tocopherol/kg Powder	^3^ mg β-Carotene/kg Powder
**Freeze-dried samples**		
^1^ Flesh	278.7 ± 11.76 ^a^	8.6 ± 1.02 ^b^
^1^ AWW	99.7 ± 16.81 ^c^	2.5 ± NA^c^
**Spray dried AWW**		
**^2^ Carrier Material**		
WPC	181.6 ± 32.24 ^b^	15.1 ± 0.23 ^a^
Lactose	131.4 ± 21.67 ^b,c^	0.0 ^c^
Acacia gum	108.1 ± 4.02 ^c^	0.7 ± NA^c^
MD 10–12 DE	95.5 ± 9.72 ^c^	0.0 ^c^
MD 17–19 DE	115.4 ± 13.39 ^c^	1.5 ± NA ^c^

Data are mean ± standard deviation on dry weight basis, all sample analysis was performed in triplicate (*n* = 3). Superscript letters within the columns for α-tocopherol and β-carotene do not differ statistically using the Tukey’s test (*p* < 0.05). ^1^ Samples were freeze dried. ^2^ Powders used were spray dried with 5% carrier loading, an inlet temperature of 140 °C, and a feed flow rate of 5.8 g/min. ^3^ Due to low β-carotene recovery, some triplicate samples only produced one reading. Hence standard deviation was not available (NA) for these.

**Table 3 foods-09-01187-t003:** Total α-tocopherol and β-carotene content of spray dried AWW without the addition of carriers.

Samples	mg α-Tocopherol/kg Powder	mg β-Carotene/kg Powder
110 °C	187.9 ± 23.09 ^a, b^	5.0 ± NA ^a^
120 °C	226.8 ± 23.44 ^a, b^	8.6 ± 4.83 ^a^
130 °C	201.0 ± 33.57 ^a, b^	3.4 ± 6.19 ^a^
140 °C	186.3 ± 44.77 ^b^	4.3 ± 3.81 ^a^
150 °C	196.9 ± 23.7 ^b^	0.4 ± 0.14 ^a^
160 °C	320.2 ± 51.09 ^a^	5.0 ± 3.83 ^a^

Data are presented as mean ± standard deviation values based on dry weight basis. Analysis of samples was performed in triplicate (*n* = 3). Superscript letters within the columns for α-tocopherol and β-carotene that are the same indicate that mean values do not differ statistically using the Tukey’s test (*p* < 0.05).

**Table 4 foods-09-01187-t004:** Trolox equivalence of additives added to pork fat determined by cupric ion reducing antioxidant capacity (CUPRAC) assay.

Additives	mg Trolox eq./100 g Powder	% *w*/*w* in Pork Fat
^1^ E316	83724 ^b^	0.04
^2^ AWW	2233 ^e^	1.50
BHT	35095 ^c^	0.10
BHA	330787 ^a^	0.01
α-Tocopherol	16745 ^d^	0.20
β-Carotene	1803 ^e^	1.86
^3^ WPC	0 ^f^	0.10

Sample analysis was performed in triplicate (*n* = 3). Superscript letters within the columns for α-tocopherol and β-carotene that are the same indicate that the mean values do not differ statistically using the Tukey’s test (*p* < 0.05). ^1^ E316 was used as baseline for all other additives. The percentage of additives added into pork was calculated as its CUPRAC equivalence to 0.04% E316. ^2^ Powder used were spray dried with 5% WPC at 160 °C inlet temperature and, feed flow rate of 5.8 g/min. ^3^ Positive control.

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
