# Peer review of "Optimising the Spray Drying of Avocado Wastewater and Use of the Powder as a Food Preservative for Preventing Lipid Peroxidation"

_foods, 2020, doi:10.3390/foods9091187_

Round 1

Reviewer 1 Report

The manuscript presents interesting results concerning microencapsulation of the ingredients contained in AWW with the intention of using the powder obtained it in the food industry. These results might be useful for practical application. However, this manuscript requires improving due to some shortcomings.

Comments that should be considered to make the manuscript suitable for publication:

L 17. Provide abbreviation for whey protein concentrate as this abbreviation was used later without explanation in line 19.

L 88. What about residual water? Can it be used as a by-product in the same way as AWW? In the introduction, it was stated that the amounts of AWW and residual water are 44.8 and 5%, respectively. What is the dry matter content of both materials?

L 192-203. Consider shifting the result concerning CUPRAC assay to Result and Discussion section.

L 228. Consider replacing “in AWW” by “from AWW” or “present in AWW”.

L 298-303. Please check the Anova results for values presented in Fig 2d. The value obtained for 5% of WPC  content is closer to the value obtained for 10% than to the value obtained for 1%. Lower case letters indicate that the values obtained for 5% and 1% belong to the same homogeneous group.

L 367-368. It is necessary to explain why spry dried AWW powder contained more antioxidants than the avocado flesh. Perhaps the results were not expressed taking into account the dry matter. The moisture content of avocado flesh is much higher than spray dried AWW powder.

L 406-409. How to explain the positive effect of the highest inlet temperature 160°C on -tocopherol content?

L 474-493. The caption of Fig 4 is missing.

L 501. Consider replacing “WPC increases MDA content” by “WPC decreases MDA content”.

I recommend to specify clearly the composition of AWW in order to show the most important ingredients. In the manuscript AWW is often equated with avocado oil (by default – diluted).

Author Response

Reviewer 1's comments

The manuscript presents interesting results concerning microencapsulation of the ingredients contained in AWW with the intention of using the powder obtained it in the food industry. These results might be useful for practical application. However, this manuscript requires improving due to some shortcomings.

Comments that should be considered to make the manuscript suitable for publication:

  1. L 17. Provide abbreviation for whey protein concentrate as this abbreviation was used later without explanation in line 19.
    • We thank the reviewer for their feedback. We have added an abbreviation for whey protein concentrate in Line 18.
  2. L 88. What about residual water? Can it be used as a by-product in the same way as AWW? In the introduction, it was stated that the amounts of AWW and residual water are 44.8 and 5%, respectively. What is the dry matter content of both materials?
    • The authors acknowledge Reviewer #1’s comment on residual water. Although residual water can also be spray dried into powder, it has very low dry matter (~3% data not shown) compared to ~12% in AWW. In addition, the volumetric output of this waste stream is significantly less than AWW hence there’s more value to valorising AWW.
  3. L 192-203. Consider shifting the result concerning CUPRAC assay to Result and Discussion section.
    • We agree with the reviewer in shifting table 1, now renamed to table 4, to line 478 in the results and discussion section.
  4. L 228. Consider replacing “in AWW” by “from AWW” or “present in AWW”.
    • “…in AWW…” has been changed to “…present in AWW…” as suggested by Reviewer #1 in line 226.
  5. L 298-303. Please check the Anova results for values presented in Fig 2d. The value obtained for 5% of WPC  content is closer to the value obtained for 10% than to the value obtained for 1%. Lower case letters indicate that the values obtained for 5% and 1% belong to the same homogeneous group.
    • Reviewer #1 is correct in identifying this issue, Figure 2D has now been corrected to show the correct statistical difference.
  6. L 367-368. It is necessary to explain why spry dried AWW powder contained more antioxidants than the avocado flesh. Perhaps the results were not expressed taking into account the dry matter. The moisture content of avocado flesh is much higher than spray dried AWW powder.
    • From our previous study, Permal, Leong Chang, Seale, Hamid and Kam [10], we found that freeze dried and spray dried AWW have higher antioxidant activity than freeze dried avocado flesh. The reason for this is still unclear. We have changed Line 362 – 363 to read, In a previous study, spray dried AWW powder was found to be rich in antioxidants, more so than freeze dried avocado flesh. This will make both samples comparable in terms of dry matter.
  7. L 406-409. How to explain the positive effect of the highest inlet temperature 160°C ona-tocopherol content?
    • We thank the reviewer for their comment. However, in this revision we have decided to split the previous “Table 3” into two separate tables now under the title “Table 2” and “Table 3”. With revised tables, our ANOVA analysis shows that inlet temperature, 160˚C, is in fact not significantly different (p > 0.05) to inlet temperature 110˚C, 120˚C, and 130˚C in terms of α-tocopherol content.
  8. L 474-493. The caption of Fig 4 is missing.
    • Reviewer #1 is correct in identifying this issue. The caption for Fig 4 has now been added in line 506 – 5011.
  9. L 501. Consider replacing “WPC increases MDA content” by “WPC decreases MDA content”.
    • The authors acknowledge Reviewer #3’s comment. However, in this specific paragraph, we are referring to WPC increasing MDA content when used as a positive control. To make this clearer, we have changed line 519-516 from “Therefore, WPC powder was used as a positive control for the TBARS test. Figure 4 confirmed that WPC increases MDA content…” to “Therefore, WPC powder that was used as a positive control for the TBARS test was shown to increase MDA content (Figure 4).”
  10. I recommend to specify clearly the composition of AWW in order to show the most important ingredients. In the manuscript AWW is often equated with avocado oil (by default – diluted).
    • We acknowledge the Reviewer’s recommendation. We have added the proximate composition of AWW (dry basis % w/w) in lines 63 – 65 to read “Proximate analysis by Permal et al. (2019) [10] showed that AWW (% dry basis w/w) is primarily composed of 53.8 ± 9.4% lipids, followed by, 22.2 ± 3.4% dietary fibres, 17.9 ± 0.6% ash, 10.3 ± 7.7% protein and 0.9 ± 3.4% available carbohydrates.”

Reviewer 2 Report

Title: Avocado wastewater powder as an effective food preservative for preventing lipid peroxidation

Authors: Rahul Permal, Wee Leong Chang, Tony Chen Brent Seale, Nazimah Hamid and Rothman Kam

Comments:

Line 76: The aims of this study were to increase and optimise AWW powder yield through spray drying and to quantify the fat-soluble antioxidants responsible for preventing lipid peroxidation.

This should be more clearly stated in the title of the publication

Most of the experiment is devoted to optimising AWW yield and quality by spray drying and less to lipid peroxidation 

Line 39: CO2   

Figure 1 does not contain enough information to illustrate the process. Quality and composition of the various fractions on each step could be included

Line 174: It is said that the pork fat was divided into 7 batches, though only 2 are explained – the rest just vaguely. Please define the variation between the different samples used in the study

Line 196: Where is Figure 4? Is it line 490? Also check the error bars and significance 

Line 200-203 – To which samples do the superscript 1, 2 and 3 refer?

Line 275: utilizing a lower inlet temperature – or – utilizing lower inlet temperatures

Line 302 – figure 2D. There must be significant difference between 1 and 5 % WPC. They are both indicated with a.

Table 3 – suggest dividing into 2 separate tables

Line 403: β-carotene

Author Response

In response to Reviewer #2.

               Comments to authors:

  1. Line 76: The aims of this study were to increase and optimise AWW powder yield through spray drying and to quantify the fat-soluble antioxidants responsible for preventing lipid peroxidation. This should be more clearly stated in the title of the publication. Most of the experiment is devoted to optimising AWW yield and quality by spray drying and less to lipid peroxidation
    • We have changed the title of the publication to “Optimising the spray drying of avocado wastewater and use of the powder as a food preservative for preventing lipid peroxidation”, we agree with Reviewer #2’s suggestion that the title needs to reflect the nature of this study.
  2. Line 39: CO2
    • We thank the reviewer for noticing this error, “CO2” on line 41 has been changed to “CO2
  3. Figure 1 does not contain enough information to illustrate the process. Quality and composition of the various fractions on each step could be included
  • We have implemented Reviewer #2’s suggestion by including a mass balance (inputs and outputs) into Figure 1. We have used an input of 1000 kg of avocado fruit as the basis. Figure 1 caption has also been re-worded to “Figure 1. Process flow diagram of a typical commercial CPAO extraction process. All the avocado waste output values from the early harvest season were obtained using an input of 1000 kg avocado fruits as the basis.”
  1. Line 174: It is said that the pork fat was divided into 7 batches, though only 2 are explained – the rest just vaguely. Please define the variation between the different samples used in the study
    • The authors have taken the reviewer’s comment into consideration. Firstly, we have decided to refer to the batches as “treatments” and secondly, we have now described each treatment thoroughly in lines 184 – 188 as advised by Reviewer #2. “Therefore, treatments 3, 4, 5 and 6 were based on the CUPRAC equivalence of each additive to E316. Treatments 3, 4, 5 and 6 contained 1.5% (w/w) AWW, 0.1% (w/w) BHT (butylated hydroxytoluene), 0.01% (w/w) BHA (butylated hydroxyanisole) and 1.86% (w/w) α-tocopherol respectively. Treatment 7 was used as a positive control and contained 0.1% (w/w) WPC.”
  2. Line 196: Where is Figure 4? Is it line 490? Also check the error bars and significance
  • Yes, this is correct. The figure caption for figure 4 had not been identified previously. This has now been amended in lines 506 – 511 to read: “TBARS values (mg MDA/kg) in cooked pork fat containing various synthetic and commercially available additives and spray dried AWW powder. Superscript letters that are the same indicate that mean values do not differ statistically when using the Tukey’s test (p < 0.05). Each analysis of sample was conducted in triplicates. Error bars represent the standard deviation of means. AWW powder used was spray dried at 160 ˚C at a flow rate of 5.8 g/min and encapsulated with 5% WPC”.

  1. Line 200-203 – To which samples do the superscript 1, 2 and 3 refer?
    • The authors acknowledge the Reviewer #2’s comments. Superscripts 1, 2 and 3 had not been added properly. This has now been corrected to, 1E316, 2AWW and 3WPC in Table 4 (Line 478)
  2. Line 275: utilizing a lower inlet temperature – or – utilizing lower inlet temperatures
    • We thank the reviewer for picking up this error, the sentence in line 273 has been changed to read: “…utilising a lower inlet temperature…”.
  3. Line 302 – figure 2D. There must be significant difference between 1 and 5 % WPC. They are both indicated with a.
    • Yes, Reviewer #2 is correct.  This has now been adjusted in Figure 2D. 1% is significantly different to 5% WPC.
  4. Table 3 – suggest dividing into 2 separate tables
    • As per Reviewer #2’s suggestion we have now separated Table 3 into 2 separate tables.
  5. Line 403: β-carotene
    • We thank the reviewer for pointing out this error. “B-carotene” in line 399 has now been changed to “β-carotene”.

Reviewer 3 Report

This is an interesting article dealing with the encapsulation of antioxidants from avocado wastewater. The experimental design is fine, the analytical methods are adequate and conveniently implemented.

I find some problems with the results and their interpretation, that should be revised:

-Firstly I would like to suggest that table 1 should be in the results section, not in the materials and methods one.

-In figure 2d, please check the subscripts  letters because I think there must be some error since the values of 1% of carrier loading seem statistically different from those of 5% and yet they have the same letter.

-Figure 2d seems to indicate a clear relationship between carrier loading and performance in the case of WPC. The comparison between different carriers has been made, apparently, at only one carrier loading (5%), is it possible that at a different carrier loading, another carrier could be better than WPC? Some kind of discussion in this sense would be interesting since otherwise it could be interpreted as an error in the experimental design.

-In table 3 the meaning of the subscripts (Shouldn't it be superscript instead of subscript?) is not clear or the conclusions are not in agreement with the results:

-line 384-385:“Freeze-dried AWW on the other hand, showed  no statistical difference compared to AWW powders encapsulated with WPC (P > 0.05)”. Does this means that 99,7 is not statically different from 181,6?

-line 386-387:“Powders dried at 110-160 ˚C without the addition of carriers exhibited a significantly higher concentration of α-tocopherol than encapsulated samples” However only the number of drying at 160 C has different letter than the others.

Line 426-427:“Higher spray drying inlet temperatures favoured increase in α-tocopherol concentration in AWW powders without carriers” Again it seems that only drying at 160 C is different from the other temperatures, and according with the table is not different from drying at 120 C since they have the same subscript letter.

Author Response

Reviewer 3's comments

This is an interesting article dealing with the encapsulation of antioxidants from avocado wastewater. The experimental design is fine, the analytical methods are adequate and conveniently implemented.

I find some problems with the results and their interpretation, that should be revised:

  1. Firstly I would like to suggest that table 1 should be in the results section, not in the materials and methods one.
    • We thank Reviewer #3 for their valuable suggestion. Table 1 is now Table 4 and moved into the results & discussion section (line 478).
  2. In figure 2d, please check the subscripts letters because I think there must be some error since the values of 1% of carrier loading seem statistically different from those of 5% and yet they have the same letter.
    • We agree with Reviewer #3. This has now been adjusted in Figure 2D. 1% is significantly different to 5% WPC.
  3. Figure 2d seems to indicate a clear relationship between carrier loading and performance in the case of WPC. The comparison between different carriers has been made, apparently, at only one carrier loading (5%), is it possible that at a different carrier loading, another carrier could be better than WPC? Some kind of discussion in this sense would be interesting since otherwise it could be interpreted as an error in the experimental design.
    • We would like to thank Reviewer #3 for the insightful suggestions. Reviewer #3 is correct that different carrier loading could perform better. From line 246 - 252, we have discussed that WPC/WPI is superior in encapsulating oily products. This is due to protein being able to form stable emulsions [25]. Based on the work reported by Du, et al. [27], 25% WPC loading of conjugated linoleic acid (CLA) gave a powder yield equivalent to that of CLA loaded with either 45% maltodextrin, 30% gum Arabic or 30% starch sodium octenyl succinate. Therefore, there is strong evidence to support that the non-protein-based carriers employed in this study will not perform better than WPC.
  4. In table 3 the meaning of the subscripts (Shouldn't it be superscript instead of subscript?) is not clear or the conclusions are not in agreement with the results:
    • The reviewer is correct, it should be superscript instead of subscript. This has now been changed in all figures and tables where appropriate.
  5. line 384-385:“Freeze-dried AWW on the other hand, showed no statistical difference compared to AWW powders encapsulated with WPC (P > 0.05)”. Does this means that 99,7 is not statically different from 181,6?
    • The reviewer is correct. Our original ANOVA test was not reflective of the true differential values. However, after splitting the table into 2 different tables (as suggested by Reviewer #2). We believe that the superscripts now show sensible significant difference. As a results, some of the discussion has been altered around this change. The changes are outlined as shown below:
      • Line 379 - 382: “…Interestingly, freeze-dried AWW showed no statistical difference to AWW powders encapsulated with lactose, acacia gum, MD 10-12 DE and MD 17-19 DE (p > 0.05). AWW encapsulated with WPC was the only spray dried sample that had significantly higher α-tocopherol content compared to freeze dried AWW.”
      • Lines 417 - 421: “…content in powders dried between 110-160˚C. Table 2 showed that WPC retained a significantly higher (p < 0.05) concentration of β-carotene (15.1 ± 0.23 mg β-carotene / kg powder) compared to AWW encapsulated with lactose, acacia gum, MD 10-12 DE and MD 17-19 DE (0.0, 0.7 ± --, 0.0, 1.5 ± -- mg β-carotene / kg powder). The results showed that WPC was the most efficient carrier for retaining both α-tocopherol and β-carotene content in AWW powder.”
  6. line 386-387:“Powders dried at 110-160 ˚C without the addition of carriers exhibited a significantly higher concentration of α-tocopherol than encapsulated samples” However only the number of drying at 160 C has different letter than the others.
    • We have taken the reviewers comments into consideration. The original table has now been separated into two and as a result, our statistical analysis has been conducted separately for AWW powders spray dried with and without carriers Therefore, the sentence in line 383 – 385 has now been reworded to read: “Powders dried at 110-160 ˚C (Table 3) without the addition of carriers exhibited a higher concentration of α-tocopherol than encapsulated samples (Table 2)”.
  7. Line 426-427:“Higher spray drying inlet temperatures favoured increase in α-tocopherol concentration in AWW powders without carriers” Again it seems that only drying at 160 C is different from the other temperatures, and according with the table is not different from drying at 120 C since they have the same subscript letter.
    • The authors agree with Reviewer #3, 160˚C is in fact not significantly different to 120˚C. The sentence in line 432 – 435 has been changed to: “Utilising the highest spray drying temperature of 160˚C did not degrade α-tocopherol concentration in AWW powders without carriers”.

Round 2

Reviewer 1 Report

In my opinion, the manuscript has been improved enough to be published in its current version